# Extracellular Vesicles Isolation from Large Volume Samples Using a Polydimethylsiloxane-Free Microfluidic Device

**DOI:** 10.3390/ijms24097971

**Published:** 2023-04-27

**Authors:** Cristina Bajo-Santos, Miks Priedols, Pauls Kaukis, Gunita Paidere, Romualds Gerulis-Bergmanis, Gatis Mozolevskis, Arturs Abols, Roberts Rimsa

**Affiliations:** 1Latvian Biomedical Research and Study Centre, Ratsupites Str. 1, k-1, LV-1067 Riga, Latvia; 2Institute of Solid-State Physics, University of Latvia, 8 Kengaraga Str., LV-1063 Riga, Latvia

**Keywords:** PDMS-free, OSTE–COC, extracellular vesicles, separation, microfluidic devices, urine, A4F

## Abstract

Extracellular vesicles (EV) have many attributes important for biomedicine; however, current EV isolation methods require long multi-step protocols that generally involve bulky equipment that cannot be easily translated to clinics. Our aim was to design a new cyclic olefin copolymer–off-stoichiometry thiol-ene (COC–OSTE) asymmetric flow field fractionation microfluidic device that could isolate EV from high-volume samples in a simple and efficient manner. We tested the device with large volumes of urine and conditioned cell media samples, and compared it with the two most commonly used EV isolation methods. Our device was able to separate particles by size and buoyancy, and the attained size distribution was significantly smaller than other methods. This would allow for targeting EV size fractions of interest in the future. However, the results were sample dependent, with some samples showing significant improvement over the current EV separation methods. We present a novel design for a COC–OSTE microfluidic device, based on bifurcating asymmetric flow field-flow fractionation (A4F) technology, which is able to isolate EV from large volume samples in a simple, continuous-flow manner. Its potential to be mass-manufactured increases the chances of implementing EV isolation in a clinical or industry-friendly setting, which requires high repeatability and throughput.

## 1. Introduction

Extracellular vesicles (EV) are membrane-bound particles secreted by all cells. When characterized by biogenesis, EV can be separated into two major types—exosomes (30–200 nm) and microvesicles (200–1000 nm) [1]. Exosomes are small EV that are formed by the inward budding of the endosomal membrane within a multivesicular body (MVB). The MVB is a specialized compartment within the cell that contains a multitude of intraluminal vesicles (ILVs) formed by the invagination of the limiting membrane. Once the MVB fuses with the plasma membrane, the ILVs are released into the extracellular environment as exosomes [1]. Microvesicles are formed by outward budding and fission of the cell membrane [2]. EV accomplish intercellular communication by transporting various proteins, nucleic acids, lipids, and metabolites [3]. The cell-specific biogenesis of EV is decided by each cell itself and represents the cell’s physiological state—including cell growth, angiogenesis, metastasis, proliferation, and therapy resistance [3]. Since this correlation exists, EV can and are used as markers for various illnesses [4]. In addition, due to the ability of EV to be readily taken up by cells, they could be utilized as a drug delivery system by encapsulating them with medication [4]. However, to fully harness the potential of EV in disease diagnostics and therapeutics, it is essential to isolate EV from various biofluids [5].

EV are isolated via various sample purification methods including ultracentrifugation (UC), [4] density gradient centrifugation, filtration, immunoisolation, and size exclusion chromatography (SEC). The most commonly used method is UC, though recently SEC has notably grown in popularity [4,6]. However, UC is still one of the golden standards for EV isolation due to the low cost of reagents and reproducibility. However, the final product always contains particles of similar density that are not EV, and the method may generate EV aggregates [7]. SEC produces samples of higher purity in comparison, since the method is based on particle size rather than density. Therefore, SEC excludes particles such as high-density lipoproteins or chylomicrons that can remain present after UC [8]. Chylomicrons are much larger in size than EV and have a different composition, consisting mainly of cholesterol, triglycerides, and apolipoproteins [9]. In theory, SEC can be used to separate chylomicrons from EV, although this would depend on the specific conditions and parameters of the SEC column [8]. However, it is important to note that chylomicrons and other particles, such as low-density lipoproteins, protein aggregates and protein complexes, can co-isolate with EV when SEC is used [7]. To minimize co-isolation of unwanted particles, it is important to carefully select the appropriate pore size for the SEC column and to optimize the conditions of the chromatography [7]. Both methods are time-consuming and difficult to automate in order to make them more industry-friendly, especially with larger volume samples such as cell media or urine [10]. Thus, other methods of EV isolation are being developed, including microfluidic devices.

Recently, microfluidics have emerged as a promising tool for the high-efficiency separation of micro- and nano-sized particles [11]. Microfluidic devices designed for small particle separation are often characterized by low sample consumption, high fabrication repeatability, reduced isolation time, low labor intensity, and high clinical reliability making them a potential substitute for UC and SEC for small particle separation [12]. Predominantly, microfluidic devices for EV isolation are of the immunoaffinity capture-based type [13], which rely on surface modifications of the channel with EV-capturing molecules [14]. Another type of microfluidic devices are physical property-based devices for separation of EV, such as acoustofluidic [15], centrifugation-based [16], and viscoelastic flow-based devices [17]. However, these methods allow manipulation only with small volumes of samples of just a few hundred microliters, or have complex fabrication and separation procedures [11]. A promising method for EV separation is A4F [18]. A4F-based devices utilize the force exerted orthogonally to the porous membrane in microfluidic channels, caused by field flow coupled with Poiseuille flow, that allows the efficient isolation of particles that have different lateral placements in the channel, caused by diffusion [19]. In the channel of the device, a laminar flow is created using the sample liquid. Then, by a transverse perpendicular flow, the sample flows towards the channel floor. Due to Brownian motion and the resulting flow profile of the channel, smaller particles elute earlier [20]. Herein, we show an improvement through a novel design and setup over the general A4F device that typically requires tedious sample concentration steps. The addition of a bifurcating flow, which forces the sample laterally closer to the membrane, ensures that smaller EV tend to stay in the fastest region of the Poiseuille flow profile, whereas all larger EV and cell debris are closer to the membrane. As a result of the bifurcating flow, this device can be run continuously, thus addressing the issue of small sample volume throughput. Therefore, devices based on this principle could be used for EV isolation from large volume samples, such as cell media from hollow fiber bioreactors to produce therapeutic EV, or urine for prostate cancer screening purposes. Proof of principle for bifurcated A4F devices has been shown to produce 3.7X enrichment of EV-sized polystyrene beads [18].

However, currently, the majority of microfluidic devices are produced from polydimethylsiloxane (PDMS). There is evidence that PDMS is problematic for devices utilizing lipophilic molecules due to absorption [21], as well as for the large-volume fabrication necessary for device repeatability [22]. Off-stoichiometry thiol-ene (OSTE) has been proposed as a potential substitute for PDMS, since it is characterized by a significantly reduced small molecule absorption compared to PDMS [23]. Moreover, its chemistry allows for easy dry bonding directly to surfaces without surface treatments and permits easy patterning [23]. Therefore, device fabrication can be scaled by reaction injection molding [24]. However, OSTE suffers from high light scattering, and complicated connections to tubing, which generally involve interference fitting that is prone to leaking [25]. Cyclic olefin copolymer (COC) is a well-suited material for microfluidic devices, due to low particle absorbance, biocompatibility, and high chemical stability [26]. However, being a thermoplastic, COC is generally not well suited for academic settings due to its complicated microstructuring and bonding procedures [27]. Additionally, its surface is non-reactive [24]. The aim of this study was to design and fabricate microfluidic devices based on A4F principles from OSTE and COC polymers suitable for mass manufacturing, for the purpose of EV isolation from large-volume samples such as urine and cell media. Furthermore, we aimed to compare the performance of the device with the current gold standard methods for EV isolation—UC and SEC.

Overall, this study demonstrates a novel bifurcated A4F setup of the microfluidic device that enables continuous flow operation, thus paving the way for large volume sample handling without additional concentration steps, which typically lead to EV loss or aggregation. The presented devices not only allow for the direct application of urine and cell culture media in the device, but they can be also mass-manufactured, thus addressing the long-standing issues of PDMS microfluidics within the EV area.

## 2. Results

### 2.1. OSTE–COC Device Fabrication and Experimental Setup

A4F is a promising method for EV separation based on size with well-established principles [19]. However, similarly to other separation methods, volumes are limited to small samples. The addition of bifurcation in the design, as shown in Figure 1a, forces the redistribution of the sample closer to the membrane, which aids further separation via downward force (due to the membrane) and Poiseuille flow, due to the microfluidic nature of the design. The operating principle of the device involves administering biofluids through the sample inlet while adding filtered phosphate-buffered saline (PBS) through the other inlet; however, other buffer solutions could be used in principle. Filtered PBS is necessary to avoid any particle generation from the buffer. Due to the separation force over the length of the device, smaller EV tend to stay in the upper half of the device, as seen in Figure 1a, and exit the device via the left outlet (L-PORT), whereas larger EV and cell debris exit the device via right outlet (R-PORT).

To address problems such as small molecule absorption and lack of scalability, as found in PDMS-based devices, we fabricated microfluidic devices from OSTE, COC, and track-etched polycarbonate (PC) membrane with 50 nm pore size (Figure 1b). OSTE was chosen due to its ease of bonding, whilst COC was the material of choice due to its high chemical stability, biocompatibility, and easy interfacing due to molded mini-luer connections. In addition, both materials are characterized by low lipophilic molecule absorption, and the use of a combination of these materials provides a scalable manufacturing process via reaction injection molding [24]. Furthermore, the combination of OSTE and COC shows better reproducibility as compared to PDMS [18]. The pore size of the membrane was chosen to prohibit EV passage through the pores while allowing the passage of smaller molecules and proteins. The upper channel dimensions were selected according to previously published results [28], and with consideration of the minimal dimension for device fabrication height being 200 μm, limited by the injection of OSTE in the channel sidewalls. For ease of fabrication, the channel cross-section aspect ratio was kept as 1:2 (height to width), which allows easy demolding from the PDMS-master molds. The bottom channel width was selected as 0.5 mm larger than the upper channel width to ensure easy alignment, given the manual nature of the task. In order to maximize the channel length, and subsequently the separation efficiency on the COC microscopy slide, a serpentine channel design was utilized. The experimental setup used for EV separation utilized a syringe pump system with different tubing used for channel inlets and outlets (Figure 1c). The use of a syringe pump system ensures pressure-independent flow for each of the inlets, given the differences in sample and PBS densities and viscosities, as well as a uniform flow rate for both inlets. The inlet and outlet tubing were made of polytetrafluoroethylene (PTFE) and polyether ether ketone (PEEK), respectively. PTFE tubing was selected due to its inherently low small molecule adsorption, whereas PEEK tubing with a 250 µm internal diameter (ID) was chosen to ensure uniform resistance in the outlets, since the L-PORT and R-PORT result in slightly different channel lengths. For the separation experiments, a flow rate of 250 μL/min for both the EV-containing sample and PBS were selected according to a previous volume flowrate parameter sweep using polystyrene beads (100 nm and 1000 nm) as a model system. This flow rate system provided the highest bead enrichment as compared to other flow rates [18].

### 2.2. OSTE–COC Device Efficacy in Particle Retrieval from Urine Samples

To evaluate the efficacy of the device in isolating EV from large volumes (>1 mL) of biofluids, we tested its performance using 20 mL of urine from 10 different healthy male donors. Urine is a heterogeneous biofluid that contains not only diluted organic and inorganic molecules, but also viruses, epithelial and blood cells, bacteria, and EV [29]. In addition, urine viscosity, acidity, and properties vary depending on multiple factors such as infections, diet, or correct hydration, making it considerably complex to compare and contrast with data across different studies [30,31].

Male urine samples (US) were selected, since this device is intended as a proof of principle for future applications such as urinary EV isolation for prostate cancer screening, for example [32,33]. EV isolation was performed simultaneously following three different EV isolation methods: UC, SEC, and the A4F microfluidics-based OSTE–COC device. After EV isolation was completed, the total amount of particles and their size distribution was assessed via nanoparticle tracking analysis (NTA) (Figure 2a–c). On average, a total number of 5.37 × 10^10^ ± 3.08 × 10^9^ particles were recovered using UC; 7.44 × 10^10^ ± 3.94 × 10^9^ using SEC, and 1.24 × 10^11^ ± 1.08 × 10^10^ using the OSTE–COC device (L+R ports) (Figure 2a). However, statistically, the OSTE–COC device only outperformed UC when the numbers of particles from both ports were combined (Figure 2a). Interestingly, R-PORT recovery (9.16 × 10^10^ ± 8.25 × 10^9^) was slightly higher than SEC and UC by itself (Figure 2a), demonstrating that total particle recovery from biofluids is better with the OSTE–COC device compared to UC. However, particle number analysis by individual sample (Figure 2b) showed that particle recovery can be donor- and method-dependent, which is in line with previously published information about US heterogeneity in terms of viscosity, particle size, density, and quantity [30,31]. These parameters can affect particle isolation by specific methods; therefore, pre-evaluation of biofluid samples (such as urine) in terms of viscosity could help to further improve the method’s reproducibility and efficiency in terms of particle recovery.

Next, we wanted to address whether the particles recovered from OSTE–COC device L and R-PORT followed any size distribution patterns, since the A4F principle is size-dependent, and whether there is a significant difference between isolation methods (Figure 2c). On average, the UC size median (with maximum and minimum range) was 164 nm (min 103.26 nm; max 312.7 nm), which was significantly larger in comparison to other methods (*p* < 0.0001). The rest of the samples showed similar total size distribution patterns with no statistical significance (SEC 147 nm (min 96.77 nm; max 283.85 nm); L-PORT 141 nm (min 89.7 nm; max 269.67 nm); R-PORT 132 nm (min 93.43 nm; max 242.45 nm). These results can be explained by particle aggregation during UC, which eventually produces larger and more heterogenous particles. This is in line with previously published data, while SEC and OSTE–COC devices are based on size distribution [34]. Particle aggregation can significantly affect the therapeutical functions of EV produced by bioreactors or the amount of EV isolated from urine for diagnostic purposes. Therefore, UC is not well-suited for these applications. Additionally, we noticed that the R-PORT recovered larger particles than the L-PORT, as expected. However, the difference was not statistically significant when comparing the L- and R-PORT between all samples (see Table 1), which suggests that there is either no size distribution between L-PORT and R-PORT or that this is due to the heterogenic nature of US.

To examine this, we compared particle size distribution between methods in each US separately (Figure 3, Appendix A). Separate analysis of each sample revealed that the OSTE–COC device could statistically significantly separate larger particles into the R-PORT compared to L-PORT in five out of ten US (3, 4, 6, 8, and 9) (see Table 1). However, in one sample (US2), particle size was increased in the L-PORT in comparison to the R-PORT. These results suggest that the A4F method’s performance could be affected by sample viscosity. The L-PORT also contained significantly smaller particles than UC in seven samples (2; 3, 4, 5, 6, 8, 9), while in comparison to SEC; the L-PORT contained significantly smaller particles in five samples (3, 4, 5, 6, 9), showing that L-PORT particle size distribution is more homogeneous than that of UC or SEC. These data shows that the A4F size distribution principle works better than UC and SEC in some samples.

Subsequently, to confirm that the isolated particles from urine contain EV, we performed a double sandwich enzyme-linked immunosorbent assay (dsELISA) on the samples using the well-known EV marker CD63 (Figure 4a,b) [4,35]. Additionally, we corroborated the purity of our sample by performing dsELISA on Calnexin, an endoplasmatic reticulum marker to show whether our sample contained intracellular debris (Figure 4c). Transmission electron microscopy was used to confirm EV morphology (TEM) (Figure 4d). Results confirmed that EV were present in all samples, based on positive CD63 and TEM. SEC and UC showed the biggest CD63 signal heterogeneity between urine EV samples in comparison to L-PORT and R-PORT; however, these significances were not noteworthy, which could be explained by US heterogeneity. On average, the highest CD63 signal was found in the R-PORT, while the lowest was reported from the L-PORT (Figure 4a). This is in agreement with particle amounts presented in Figure 2a, therefore confirming that the OSTE–COC device has overall better particle recovery of urinary EV based on dsELISA when combining the L- and R-PORT. Furthermore, we evaluated CD63 signal variation between methods in each sample separately (Figure 4b). Results showed a similar scenario to the total particle amount in Figure 2b, pointing out the heterogeneity between samples and methods. Interestingly, the CD63 signal does not represent the particle amount in every sample. For example, US6 presents the highest CD63 expression in SEC (Figure 4b), but its particle amount in SEC is lower in comparison to other samples (Figure 2b). However, this can be explained by the fact that CD63 is not a unique marker for EV, and that urinary EV markers vary significantly between donors, storage conditions, or even sample collection time [4,30,36]. To confirm the purity of our isolated particle cohort, we tested samples for Calnexin expression (Figure 4c). A mild positive Calnexin expression was detected in samples isolated using the three methods, with the lowest expression identified in the L-PORT sample and the highest in SEC and R-PORT (Figure 4c). While this could be an indicator of cell debris presence, the expression was insignificant when compared to cell expression (+) or CD63-EV marker. Therefore, we can conclude that isolated samples are at least enriched in urinary EV. In addition, representative TEM pictures were taken of each method from three different US (1, 8, 9) (Figure 4d). Pictures confirmed the cup-shaped morphology of EV typical for TEM. UC samples also showed more heterogeneity and larger particles on TEM, while SEC showed more heterogeneous samples in comparison to L-PORT and R-PORT, as shown in Figure 3. Overall, these results confirmed that OSTE–COC devices have better EV recovery from urine when compared to SEC and UC. However, particle size distribution between the L-PORT and R-PORT is US dependent, and flow rates of sample and buffer need to be optimized based on sample parameters, such as viscosity, to implement the full potential of the A4F principle.

### 2.3. OSTE–COC Device Efficacy in Particle Retrieval from Cell Media Samples

To evaluate the applicability of the OSTE–COC device to EV research settings, we tested it using conditioned cell culture media. Conditioned cell media is a complex liquid consisting of various biomolecules and biochemical components, which are generally collected in large volumes and require several steps and time-consuming processes to isolate EV, which can jeopardize EV quality. Therefore, we also tested and compared the efficacy of OSTE–COC versus UC and SEC in isolating particles from 20 mL of cell media from two prostate cancer cell line cultures: prostate cancer (PC3) and lymph node carcinoma of the prostate (LNCaP). These stable cell lines were used for these tests since they are most often used for prostate cancer EV research [37], and they have been shown to express high levels of CD63 in EV [38]. We compared the number of particles recovered and their size distribution by NTA (Figure 5a–d). EV presence and purity was confirmed by CD63 and Calnexin dsELISA tests (Figure 5e–g). EV morphology was confirmed by TEM (Figure 5h).

In a similar manner to the findings from the US, the combination of L-PORT and R-PORT resulted in increased particle amounts in PC3 (Figure 5a) and LNCaP (Figure 5b) cell media compared to UC and SEC. However, differences between methods were greater in LNCaP cells, indicating that LNCaP cells produce more particles that are lost during UC or SEC isolation. The size distribution graph revealed that the OSTE–COC device’s L-PORT had significantly smaller particles and a more homogeneous distribution than UC, SEC, and the R-PORT in PC3 cell media (Figure 5c), indicating that the A4F principle worked well for this sample. However, no statistical significance was observed in LNCaP cell media, except between R-PORT and UC (Figure 5d), indicating that A4F principle performance and EV isolation overall depend on various biofluid parameters, even in cell media.

The dsELISA results of CD63 signal from cell line media showed signal heterogeneity, depending on the cell media and method used (Figure 5e,f), similar to the particle amount findings, thus suggesting that at least a majority of the isolated particles are EV. However, similar to US, we conclude that flow rate optimizations are necessary for each sample separately based on sample viscosity to fully implement the A4F principle and shift more EV particle collection into the L-PORT. Moreover, the EV samples obtained from conditioned cell media returned negative results for Calnexin, indicating the purity of the EV isolated sample in terms of cell debris (Figure 5g). Typical TEM EV cup-shaped morphology was confirmed from Figure 5h. UC and SEC samples showed larger particles and more heterogeneous samples. However, the TEM images of L- and R-PORT demonstrated high protein contamination in samples from cell media, making it difficult to acquire high-quality TEM pictures (Figure 5h), and suggesting that the current setup cannot purify EV from smaller protein complexes found in cell media. This can be optimized in future by testing different pore size membranes at the bottom of the device in order to remove these particles during buffer crossflow.

In summary, these results show that the OSTE–COC device has better particle recovery from both urine and cell media samples compared to UC and SEC based on NTA, and the majority of these particles are EV, based on dsELISA and TEM. However, there was sample- and method-dependent variation in EV recovery and size distribution, which can be affected by sample physical characteristics such as viscosity and density. Additionally, to fully implement bifurcation of the A4F principle and completely shift EV recovery from R-PORT to L-PORT, flow rate optimizations for both sample and buffer are necessary for each sample, and previously selected flow rates from a parameter sweep using polystyrene beads (100 nm and 1000 nm) as a model system are not representative. While this may seem challenging, we showed that sample heterogeneity also affects other methods, making this a concern that needs to be addressed for future industrial EV isolation purposes.

Finally, further optimization in device design and setup is necessary to achieve even higher EV-sample uniformity and purity. Some of the device-altering parameters are the relationship between channel height and total meandering length, while a study of sample flow to buffer flow ratio from the experimental setup would be beneficial to understand its effect on EV sample size distribution and homogeneity.

## 3. Discussion

Current gold standard methods for EV separation and extraction from biological fluids, namely, UC and SEC, are highly labor intensive and time-consuming with high variability [11]. However, for therapeutics or applications such as drug delivery and cosmetics, large numbers of EV are necessary, inevitably involving substantial volumes of samples [39,40]. Furthermore, given the scarce amounts of EV recovered from biological samples, large sample volumes are required to study minimally invasive diagnostics, for example urine [41].

One of the promises of microfluidic and particularly lab-on-chip technology has been the ability to reduce the biological sample preparation for increased throughput [22,42]. However, currently, the majority of developed microfluidic devices are designed for handling small sample volumes, and their performance is often limited by the eventual clogging of the device [20]. Hence, the device presented herein is of particular interest since it can continuously process 20 mL of US with little input from the user in the form of separated EV collection. Furthermore, the devices shown here are also compatible with large volume manufacturing via the reaction injection molding process of OSTE and COC substrates, thus addressing the idea that, for truly high throughput, device automation and subsequently large volume manufacturing of devices are necessary, which PDMS devices cannot address [43]. Mass manufacturing of an EV isolation device can improve reproducibility compared to UC and SEC techniques, which rely on manual steps and can introduce variability. Standardizing the device’s performance reduces variability and improves reproducibility, which is essential for applications such as EV-based therapeutics or diagnostics [4]. Our device has the added advantage of processing large sample volumes with minimal user input, and its compact size makes it convenient for use in industrial or clinical settings. However, further research is required to evaluate the device’s performance consistency across varied sample properties, such as different viscosity and density of various biological samples.

The size distribution produced by our device is smaller than that of the UC method (Figure 2c and Figure 5c,d), which is a widely used method for the isolation of EV from large volume samples [4]. A better alternative to our device with a similar size distribution would be SEC. SEC for EV isolation offers high purity and specificity, as it effectively removes smaller contaminants and aggregates from the sample [7]. Additionally, SEC is a gentle method that does not require harsh chemicals or high-speed centrifugation, which can damage or alter the EV [8]. However, SEC also has some limitations. One of the main drawbacks is its low yield, as parts of EV can be lost during the separation process due to binding to the column matrix [7]. Moreover, both SEC and UC methods are time-consuming and require specialized equipment, making them less suitable for high-throughput applications [12]. Therefore, it is crucial to explore alternative methods for EV separation and extraction that can avoid the issue of aggregation, ensure the optimal recovery rate, and be gentle to the particles intended for downstream applications.

Additionally, one of the major limitations while isolating EV is co-isolation with similar size-nature particles, which affects the purity of the sample [8]. Previously, optimized dsELISA for the recognition of EV [44] was tested to assess the expression of well-known positive and negative EV markers CD63 and Calnexin, respectively [4]. Our results confirmed that at least part of the isolated particles are EV and samples do not contain cellular debris (Figure 4a–c and Figure 5e–g). Additionally, our device outperformed UC and SEC in terms of CD63 positive particle isolation with both ports combined. Nevertheless, since every EV isolation method relies on different parameters, it is unavoidable that a specific subset of EV particles will be enriched; therefore, larger marker testing [4] could reveal possible reasons behind the huge heterogeneity between US and cell media samples.

It is also evident that the physical properties of the sample also play a significant role in the separation [14]. Subsequently, ten different US provide a good example of real-world application efficacy for this technology in comparison to UC and SEC.

To improve device performance, an upstream sample characterization is likely necessary to quantify sample density and viscosity to adjust the subsequent flow parameters [45]. These parameters were kept the same here for consistency across sample preparation. Considering that sample flow can be continuously adjusted, this device principle can also be combined with high-resolution flow cytometry and pre-developed algorithms to adjust flow rates of sample and buffer in real-time to enhance EV isolation for each sample [46], overcoming the sample heterogeneity challenge. Thus, this could also allow for automation of EV isolation from large volumes of highly heterogeneous samples in the future.

In comparison to UC, if high concentrations of EV in low volume samples are desired, concentration of the EV samples isolated from the OSTE–COC device presented here can be achieved using common filtration tubes, tangential field flow (TFF), or microfluidic twisting [47]. Additionally, concentration tubes with larger pore sizes or TFF can be used to further purify samples isolated by the A4F principle, therefore even further automating EV isolation.

The overall advantage of our approach to EV isolation, in comparison to UC and SEC, is that our method allows for continuous EV isolation, does not require sample pretreatment like SEC, and allows mass manufacturing of such devices, which could significantly increase the reproducibility of EV isolation from samples due to the reduced variability of EV isolation device manufacturing. Moreover, unlike ultrafiltration, which relies on the use of membranes to isolate EV [48], our method does not involve such a step. With ultrafiltration, a portion of EV tend to adhere to the membrane, even when low protein-affinity materials are used. The process involves using centrifugation, pressure, or vacuum to force the sample through the membrane. However, the concentration of contaminants and protein molecules in the sample tend to clog the membrane pores during the concentration step, which slows the process and may result in the loss of the target material [48]. Furthermore, the potential impact of pressure, vacuum, and contact with the membrane on the deformation of EV can affect EV functionality. Finally, such a microfluidic module could, in principle, be integrated into different devices such as hollow fiber cell bioreactor cartridges, high-resolution flow cytometers, and others to make EV applications more industry friendly.

## 4. Materials and Methods

### 4.1. Microfluidic Device Fabrication from OSTE and COC Polymers

The microfluidic device was fabricated and tested as described in our previous research [18]. In short, the microfluidic chip was fabricated using the soft lithography process. A double negative mold was fabricated via liquid-crystal display (LCD) 3-dimensional (3D) printing (Zortrax Inkspire, Olsztyn, Poland) and pretreated as mentioned in [49]. A mixture of PDMS oligomer and cross-linking polymer PDMS (PPS, Chanhassen, MN, USA, QSUL 216) with weight ratio 10:1 was degassed and cast onto the 3D printed mold and cured at 60 °C for 12 h. The negative PDMS mold was removed from the 3D printed mold and brought in contact with an oxygen plasma (PVA TePla AG, Wettenberg, Germany, GIGAbatch 360M)-treated COC luer slide (microfluidic ChipShop, Jena, Germany). A mixture of OSTE 322 (Mercene Labs, Stockholm, Sweden) part A and part B was used to fill in the PDMS cavities via a pressure system (Elveflow microfluidics, Paris, France, OB1 MK3+) and cured with an 850 mJ/cm^2^ dose of ultraviolet (UV) light. Cured OSTE was then removed from the PDMS mold, brought in contact with a track-etched polycarbonate (PC) membrane with 50 nm pores and 11.8% pore density (it4ip, Louvain-La-Neuve, Belgium), and cured overnight at 60 °C. Similarly, an oxygen plasma-treated COC slide was brought in contact with a PDMS mold. OSTE was used to fill in the PDMS cavities and UV cured with a 1100 mJ/cm^2^ dose. Cured OSTE was then brought in contact with the other side of the PC membrane and cured overnight at 60 °C. Bonding performance for the chips was tested by passing 20 nm filtered deionized water through the channels using a pressure system with 30 mbar pressure for 1 min, followed by 100 mbar pressure for 10 s. During the tests, the device was carefully examined for leaks. If no leakage was observed, the device was used for further experiments. Channel dimensions were selected to minimize the aspect ratio and thus improve the demolding and device yield with regard to channel width, whereas channel height was set at 0.5 mm, as per the work of Sitar et al. [28]. Overall channel length was selected as the longest continuous channel that can be fabricated onto the COC microscopy slide format.

### 4.2. Urine Sample Collection and Processing

Approximately 100 mL of morning urine samples (US) were collected from 10 healthy male donors aged 19 to 32 years old (Mean: 24.6 years) following pre-established biobanking procedures by the Latvian Genome Center [50]. US were processed within 2 h of collection as previously described [51]. Briefly, each sample was centrifuged at 2000× *g* for 15 min at room temperature (RT) and the supernatant was collected. To get rid of large particles, samples were then centrifuged at 10,000× *g* for 30 min at RT. After centrifugation, the supernatant was collected, and the pellet was discarded. For each EV isolation method, 20 mL of supernatant was used.

The study was conducted according to the Declaration of Helsinki. Samples were collected after obtaining donors’ prior informed written consent. The study protocol was approved by the Latvian University Life and Medical Science Research Ethics Committee (decision No. 71-35/54).

### 4.3. Cell Lines

Two prostate cancer cell lines were selected for EV isolation: prostate cancer (PC3) and lymph node carcinoma of the prostate (LNCaP). Both cell cultures were purchased from the American Type Culture Collection (ATCC, Manassas, VA, USA) and cultured in a 5% CO_2_ humidified environment at 37 °C.

PC3 cells were cultivated in high glucose DMEM/F-12 (Thermo Fisher Scientific, Waltham, MA, USA, 31330-095) supplemented with 10% heat-inactivated fetal bovine serum (iFBS) (Sigma-Aldrich, St. Louis, MO, USA, #F7524) and 50 μg/mL Primocin^®^ (Invivogen, San Diego, CA, USA, ant-pm-2). LNCaP cells were grown in Roswell Park Memorial Institute Medium (RPMI) (Thermo Fisher Scientific, 52400-205) supplemented with 10% iFBS and 50 μg/mL Primocin^®^.

To collect EV, cells were cultured in 20 T175 flasks per cell line until they reached 80% confluence. The cells were then collected, counted, and resuspended in 100 mL of serum-free media supplemented with 2% B-27 serum substitute supplement (Thermo Fisher Scientific, #17504044). A total of 108.5 million PC3 and 99 million LNCaP cells were resuspended.

After resuspension, each cell line was cultured in a separate single T175 suspension flask for 48 h. Subsequently, media was collected and centrifuged at 300× *g* for 5 min at RT. The supernatant was then collected and centrifuged again at 3000× *g* for 30 min at +4 °C. After centrifugation, the supernatant was collected, and the pellet was discarded. 20 mL of supernatant was used for each EV isolation method.

### 4.4. EV Isolation via UC, SEC, and OSTE–COC Device

EV were isolated from previously centrifuged samples (as described in sections “Cell Cultures” and “Urine sample collection and processing”) using three different isolation methods, UC, SEC, and the OSTE–COC device, to compare the recovery rate, purity, and particle size distribution of each method. To reduce the variability between the three methods, they were performed simultaneously after the centrifugation of each sample.

UC: 20 mL of urine or conditioned cell media was centrifuged at 100,000× *g* for 70 min at +4 °C using an ultracentrifuge (Beckman Coulter, Brea, CA, USA, Optima L100XP) with a fixed angle Type 70 Ti rotor (Beckman Coulter, 337922). After discarding the supernatant, the pellet was resuspended in 20 mL of 20 nm filtered phosphate-buffered saline (PBS) and centrifuged again at 100,000× *g* for 70 min at +4 °C using the same rotor. The supernatant was then discarded, and the pellet was resuspended in 12 mL of 20 nm filtered PBS, prior to centrifuging at 100,000× *g* for 70 min at +4 °C using an SW40 Ti rotor (Beckman Coulter, 331301). Finally, the pellet was resuspended in 100 μL of 20 nm filtered PBS. The resuspended sample was aliquoted and stored at −80 °C for further use.

SEC: 20 mL of urine or conditioned cell media was concentrated to 500 μL using 100 kDa molecular weight cut-off (MWCO) centrifugal filter units (Merck Millipore, Burlington, MA, USA, UFC910024) at 3000× *g* at +4 °C for approximately 15 min. Next, the concentrate was transferred onto a qEVoriginal/35 nm column (Izon, Christchurch, New Zealand, SP5), and a total of 15 fractions (0.5 mL each) were collected. The fractions containing EV were determined with a Zetasizer Nano ZS (Malvern Panalytical, Malvern, UK) instrument. Fractions with an attenuation index lower than 11 and with a proportion of more than 30% of the total particles being bigger than 40 nm were selected as EV-containing fractions. Selected fractions were pooled together, followed by concentration up to 100 μL using 3 kDa MWCO centrifugal filter units (Merck Millipore, UFC200324) at 14,000× *g* at +4 °C. The concentrated sample was then aliquoted and stored at −80 °C for further use.

EV isolation by the OSTE–COC device: For the experimental setup, a syringe pump system (DK Infusetek, Shanghai, China, ISPLab02) was used to ensure a continuous flow rate. To ensure uniform resistance in channel outlets, 800 µm inner diameter (ID) polytetrafluoroethylene (PTFE) tubing (Darwin microfluidics, BL-PTFE-1608-20) was connected to the inlets, while 250 µm ID polyether ether ketone (PEEK) tubing (Darwin microfluidics, CIL-1581) was connected to the outlets. For separation experiments, a flow rate of 250 µL/min for both sample and PBS buffer was used, which was previously determined as the optimal flow rate to separate 100 nm polystyrene beads (Invitrogen, #F8803) from 1000 nm beads (Invitrogen, #F13083) [18]. Prior to EV isolation, the device was washed with 2 mL of 20 nm filtered 3% H_2_O_2_ and 2 mL of 20 nm filtered 70% ethanol to disinfect the channels, 4 mL of 20 nm filtered PBS, and left overnight filled with PBS. The following day, the device was washed again using 4 mL of 20 nm filtered PBS to remove any air bubbles that may have precipitated from the buffer. To isolate EV using the device, 20 mL of urine or conditioned cell media sample was administered through the sample inlet, while 20 mL of 20 nm filtered PBS was inserted through the buffer inlet at the same flow rate. The flow-through was collected from the left outlet of the OSTE–COC device (L-PORT) and the right outlet of the OSTE–COC device (R-PORT) separately and concentrated using 100 kDa MWCO centrifugal filter units at 3000× *g* at 4 °C until the sample volume was reduced to 100 µL. The sample was aliquoted and stored at −80 °C for further use. A PBS sample was collected before the sample was administered into the device and used as control to ensure that any particles detected did not come from the device or buffer.

### 4.5. EV Characterization

EV purity, quantity, and morphology were analyzed using transmission electron microscopy (TEM), nanoparticle tracking analysis (NTA), and high affinity T-cell membrane protein 4 (TIM-4) double sandwich enzyme-linked immunosorbent assay (dsELISA).

To determine particle size distribution and amount, samples were diluted 1:1000 in 20 nm filtered PBS and analyzed by NTA using NanoSight (Malvern Panalytical, NS300) with a scientific metal–oxide–semiconductor camera and a green (532 nm) laser. Five measurements of 60 s were made using camera level 14, screen gain 1, and the recordings were analyzed using NanoSight NTA software v3.4 Build 3.4.003 with a detection threshold of 5. Before EV measurements, the instrument was calibrated with 200 nm polystyrene latex standard beads (Malvern Panalytical, NTA4089) diluted 1:50 in 20 nm filtered Milli-Q water. The diluted beads were measured and assessed following manufacturer’s instructions.

EV presence in samples was confirmed by the EV surface marker CD63, and the absence of cells or cell particles was confirmed by the cell marker protein Calnexin. This was determined by performing a high affinity TIM-4 dsELISA as mentioned in [44]. The dsELISA plate (Greiner Bio-One, Kremsmunster, Austria, 655-001) was coated with TIM4-Fc protein (Adipogen LifeSciences, San Diego, CA, USA, AG-40B-0180B-C010) by adding 100 μL of 0.6 μg/mL TIM4-Fc to each well and letting it bind overnight at +4 °C, shaking at 200 rpm. The following day, all coated wells were washed with washing buffer (2 mM CaCl_2_, 150 mM NaCl, 20 mM Tris (Hydroxymethyl)Aminomethane, pH 7.5, 0.05% Tween-20 (Sigma-Aldrich, 9005-64-5)) by adding 200 μL to each well and placed on a shaker for 5 min at 200 rpm at RT. This step was repeated twice before blocking each well for 1 h at RT by using 200 μL blocking buffer (washing buffer + 1% (*w*/*v*) bovine serum albumin (BSA) (Sigma-Aldrich, A7906-100G)) per well. Afterward, wells were washed twice again using 200 μL of washing buffer per well and by placing the plate onto a shaker for 5 min at 200 rpm at RT. After washing, EV samples were diluted 1:100 of the total volume using washing buffer, and 100 μL of diluted EV sample was loaded into each well and left to bind for 90 min at RT while shaking at 200 rpm. After binding, each well was washed four times by adding 200 μL of washing buffer to each well, and by placing the plate onto a shaker for 5 min at 200 rpm at RT. Next, EV were confirmed by attaching CD63 primary antibody (Santa Cruz Biotechnology, Dallas, TX, USA, sc-5275) to the EV by loading 100 μL per well of 0.67 μg/mL primary antibody diluted in washing buffer and incubating at RT for 2 h on a shaker at 200 rpm. For confirming cells or cell particles, Calnexin primary antibody was used instead (1:250, Santa Cruz, sc-80645). Afterward, each well was washed four times by adding 200 μL of washing buffer to each well, and by placing the plate onto a shaker for 5 min at 200 rpm at RT. Subsequently, 100 μL of 0.40 μg/mL mouse immunoglobulin kappa binding protein conjugated to horseradish peroxidase (m-IgGκ BP-HRP) (Santa Cruz Biotechnology, sc-516102) diluted in washing buffer was loaded per well and incubated for 1 h at RT while shaking at 200 rpm. After binding, each well was washed four times by adding 200 μL of washing buffer to each well and by placing the plate onto a shaker for 5 min at 200 rpm at RT. After the final washing step, 100 μL of 3,3′,5,5′-tetramethylbenzidine (TMB) (Sigma-Aldrich, T8665-100ML) was added to each well to start the reaction and left incubating at RT for 30 min. To stop the reaction, 100 μL of 1M H_2_SO_4_ was added to each well. Absorbance was measured at 450 nm by using a spectrophotometer (BioTek, Winooski, VT, USA, μQant) and obtaining the data through Gen5 2.0 software. To ensure data credibility, a blank control was made. Additionally, to ensure there is no unspecific binding of m-IgGκ BP-HRP, a non-primary antibody control was made. Duplicates for each sample and control were made.

To determine EV morphology, samples were observed using JEM-1230 TEM (JEOL, Peabody, MA, USA). 10 μL of each sample was attached to a carbon-coated 300-mesh grid and then incubated with 1% (*w*/*v*) uranyl formate. The mesh was placed under the TEM, and multiple pictures of different points were taken.

### 4.6. Statistical Analysis

Figures and working principle of the model were created using InkScape 1.1.2 (InkScape project, Boston, MA, USA). Graphs and statistical analysis were performed using GraphPad Prism 8.0 (GraphPad, San Diego, CA, USA). Friedman’s test with Dunn’s multiple comparison correction was performed to analyze distribution differences among groups using UC measurements as reference. *p*-values ≤ 0.05 were considered significant.

## 5. Conclusions

We have presented a novel microfluidic device that utilizes the bifurcated A4F separation principle for continuous EV separation from large volumes (20 mL) of US and conditioned cell media. The device shows a considerable shift in size distribution of the EV, suggesting that the bifurcation indeed forces smaller EV through one of the device outlets more preferentially, thus demonstrating the potential of continuous size-based EV sorting from real US or conditioned cell media. We envisage that this technology has the potential to disrupt the EV separation and purification market due to its simplicity from the user’s perspective and compatibility with large-scale volume manufacturing.

## Figures and Tables

**Figure 1 ijms-24-07971-f001:**
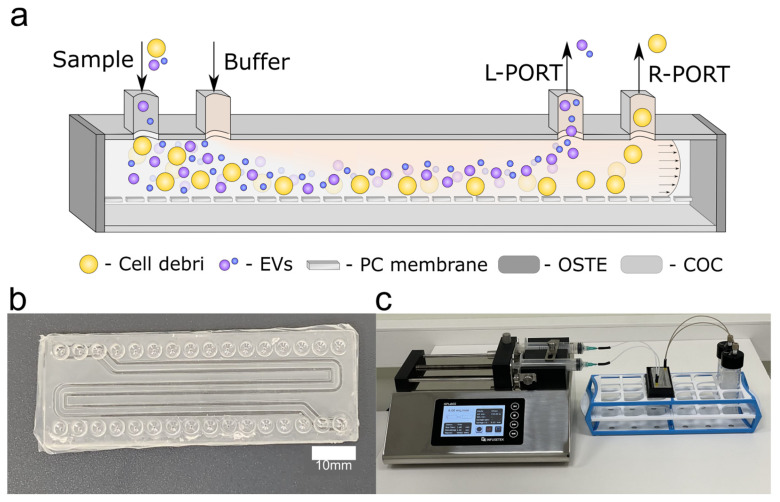
OSTE–COC device operation principle and experimental setup (**a**) schematic of device principle: sample and buffer are added in parallel inlets, at a constant flow rate of 250 μL/min. The fraction containing smaller particles is collected through the L-PORT outlet while bigger particles are recovered through the R-PORT; (**b**) fabricated OSTE–COC device with serpentine channel. Scale bar: 10 mm; (**c**) picture of device’s experimental set-up. On the left is the syringe pump, in the middle is the OSTE–COC chip, and on the right is the recovery station.

**Figure 2 ijms-24-07971-f002:**
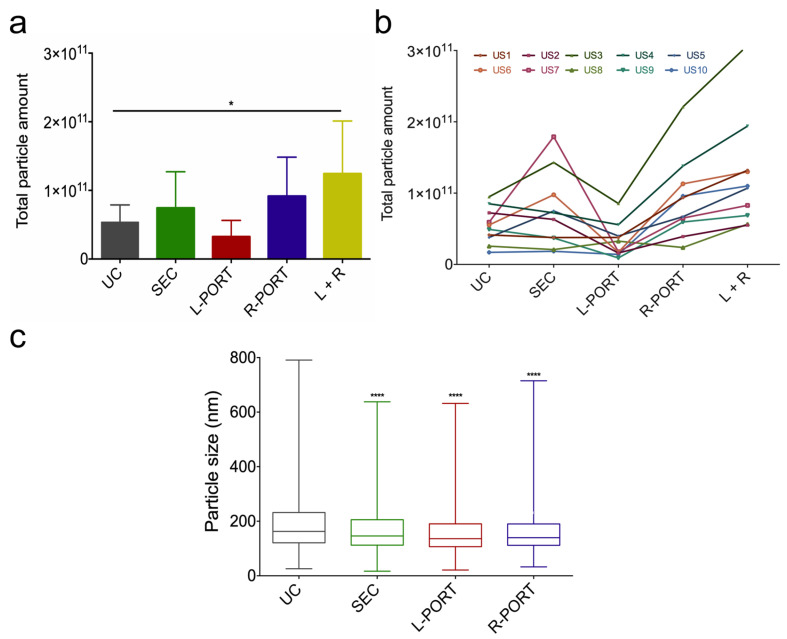
Comparison of particle recovery and size distribution by UC, SEC and the OSTE–COC device from 10 US. (**a**) Particle amount recovered from US by each of the isolation methods evaluated by NTA (Mean ± SD). * *p* < 0.05; (**b**) individual particle amount recovered per US for each isolation method evaluated by NTA; (**c**) boxplots showing average particle size distribution among all US evaluated by NTA. Whiskers show minimum and maximum. *p*-values derived from comparison to UC, **** *p* < 0.0001. US1–10—individual urine samples from ten donors.

**Figure 3 ijms-24-07971-f003:**
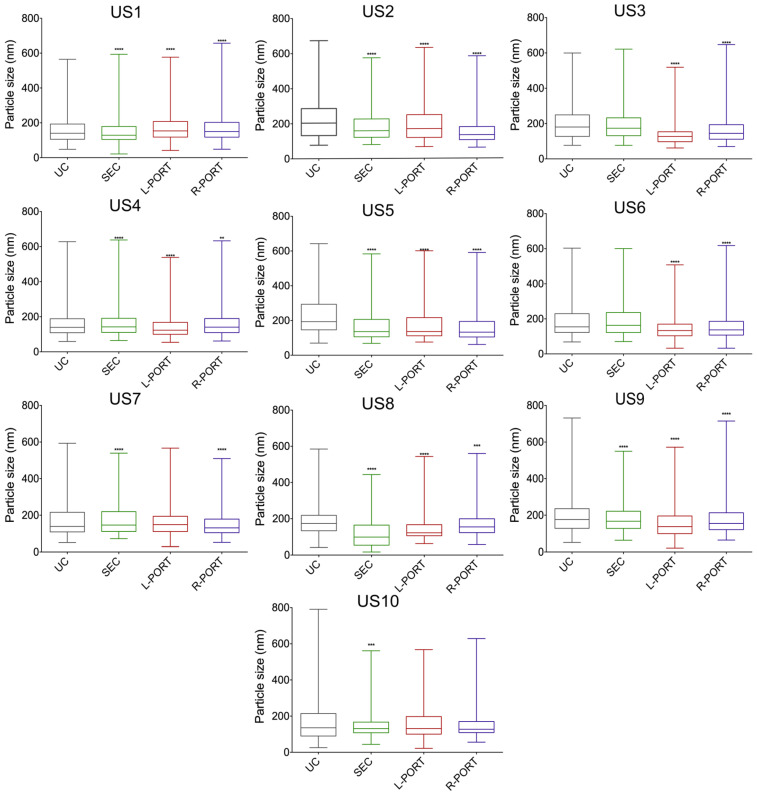
Size distribution of all individual US compared by isolation method. Boxplots with whiskers showing minimum and maximum values. p-values derived from comparison to UC. ** *p*-value < 0.01; *** *p*-value < 0.001; **** *p*-value < 0.0001. US1–10—individual urine samples from ten donors. Data generated by NTA.

**Figure 4 ijms-24-07971-f004:**
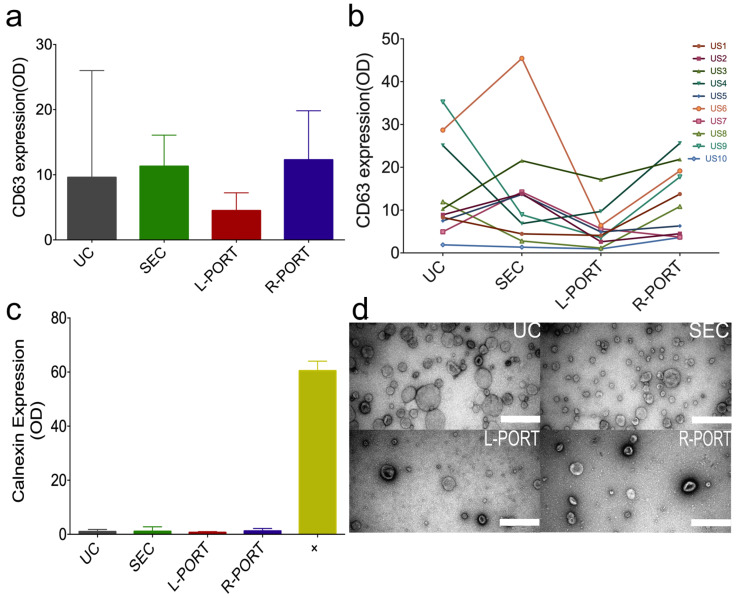
Characterization of urinary particles by isolation method. (**a**) Median and range of average CD63 amount evaluated by dsELISA for each isolation method from all US; (**b**) dynamics of CD63 amount evaluated by dsELISA for each sample across different EV isolation methods; (**c**) median and range of average Calnexin amount evaluated by dsELISA for each isolation method from all US. +—Positive control (PC3 cells); (**d**) representative TEM pictures per isolation method from US9. Scale bar: 500 µm. UC and SEC samples were used undiluted for visualization, while L-PORT and R-PORT samples were diluted 1:4. US1–10—individual urine samples from ten donors.

**Figure 5 ijms-24-07971-f005:**
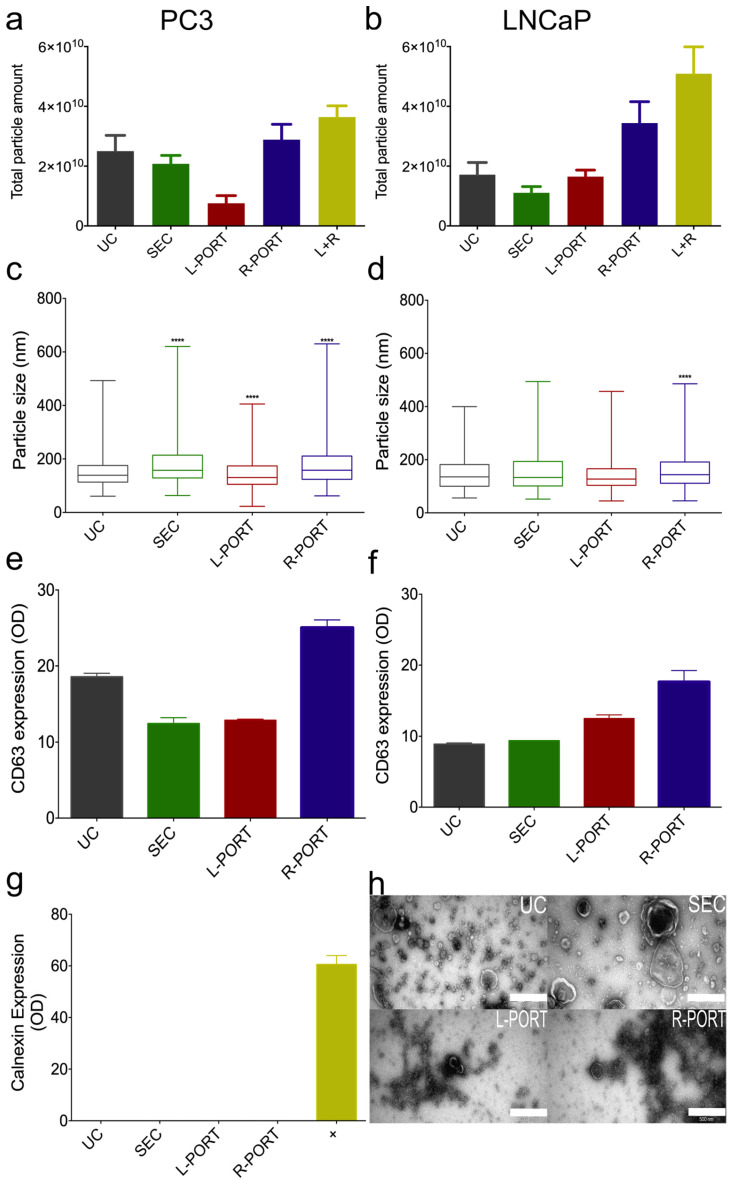
Characterization of cell particles by isolation method. (**a**) Particle amount recovered from PC3 cell media by each of the isolation methods evaluated by NTA (Mean ± SD); (**b**) particle amount recovered from LNCaP cell media by each of the isolation methods and evaluated by NTA (Mean ± SD); (**c**) median particle size distribution from PC3 cultures with range by NTA; (**d**) median particle size distribution from LNCaP cultures with range by NTA; (**e**) median and range of PC3 cell line-derived EV surface CD63 amount evaluated by dsELISA for each of the isolation methods; (**f**) median and range of LNCaP-derived EV CD63 expression dsELISA by each of the isolation methods; (**g**) median and range of average Calnexin expression dsELISA by each of the isolation methods from LNCaP and PC3 EV combined. +: Positive control isPC3 cells; (**h**) representative TEM pictures of isolated particles per isolation method for EV isolated from LNCaP. Scale bar: 500 µm, UC and SEC samples were used undiluted for visualization, while L-PORT and R-PORT samples were diluted 1:4. **** *p*-value < 0.0001.

**Table 1 ijms-24-07971-t001:** Adjusted *p*-values of multiple rank comparisons among particle size distribution from different isolation methods and outlet ports per US. #—particle size statistically significant increase in SEC, $—particle size statistically significant increase in R-PORT, *—particle size statistically significant increase in L-PORT, ns—not significant.

Sample	UC vs. SEC	UC vs. L-PORT	UC vs. R-PORT	SEC vs. L-PORT	SEC vs. R-PORT	L-PORT vs. R-PORT
All	<0.0001	<0.0001	<0.0001	<0.0001	<0.0001	ns
US1	<0.0001	* <0.0001	$ <0.0001	<0.0001	$ <0.0001	ns
US2	<0.0001	<0.0001	<0.0001	ns	<0.0001	* <0.0001
US3	ns	<0.0001	<0.0001	<0.0001	<0.0001	<0.0001
US4	# <0.0001	<0.0001	$ 0.0053	<0.0001	0.0053	<0.0001
US5	<0.0001	<0.0001	<0.0001	<0.0001	<0.0001	ns
US6	ns	<0.0001	<0.0001	<0.0001	<0.0001	0.0044
US7	<0.0001	ns	<0.0001	ns	<0.0001	ns
US8	<0.0001	<0.0001	0.0007	* 0.0003	$ <0.0001	<0.0001
US9	<0.0001	<0.0001	<0.0001	<0.0001	$ <0.0001	<0.0001
US10	0.0019	ns	ns	ns	ns	ns

## Data Availability

Not applicable.

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
