# Peer review of "Extracellular Vesicles Isolation from Large Volume Samples Using a Polydimethylsiloxane-Free Microfluidic Device"

_ijms, 2023, doi:10.3390/ijms24097971_

Round 1
Reviewer 1 Report
This study is interesting with clinical significance of EV. EV isolation methods require long multi-step protocols generally involving bulky equipment that cannot be easily translated to clinics. The authors put forward a new point of view to solve this problem. The followings are comments to the authors.
1.Please define the abbreviations in the text when used for the first time. For example, in line 13 “COC-OSTE” .
2. Figure 1a showed R-PORT recovery was slightly higher than SEC and UC by itself. But in TEM pictures (Figure 4d) ,the number of EV is less in R-PORT group than in SEC and UC groups. The same thing happened in Figure 5a and Figure 5h. Please explain that.
3.I suggest selecting multiple markers for identification of EV, such as CD9,CD81and so on.
4.The discussion and conclusion can be improved. These kinds of studies have limitations. Hence, the author should have stated the potential limitations and suggested what could be done the next step in this area of research.
Author Response
We would like to express our gratitude for your effort in reviewing our manuscript and offering insightful feedback. We have carefully considered all of your comments and suggestions, and as a result, we have made significant revisions to the manuscript to improve its quality and accuracy. We sincerely appreciate your valuable time and contribution towards improving our work. We are confident that these revisions have addressed your concerns and improved the overall clarity of the paper. We hope that you will now find the revised version of the manuscript acceptable for publication and we look forward to hearing your feedback.

Reviewer 2 Report
The title of the manuscript is remarkable. English language is easy to understand. Figures and tables have good quality.The citaions in the main text have some problems like multiple and middle sentence references. Both part "Results" and "Discussion" have some major problems (but can be reformed)
1. Page 1, line 28-31 and line 33-36
Why the sentences in this part have no reference?
2. Page 1, line 42 does not have reference. Why?
3. Page 1, line 45
Why this part has multiple reference?
4. Part 2, line 58
Please reconsider multiple reference here
5. Page 2, line 61
Please reconsider middle-sentence reference here
6. Page 2, line 64-65
Please insert Figure 1a right after you mention its name for the first time
7. Page 2, line 84, 88 and 94
Please reform all multiple references here?
8. In Page 3, part "2.2. Urine sample collection and processing" there are two problems:
First: the authors have mentioned that "Approximately 100 ml of morning urine samples (US) were collected from 10 healthy 134 male donors aged 19 to 32 years old
(Mean: 24.6 years)." but they have not mentioned what was their criteria for evaluation the health of urine donors. Please determine criteium that you have assessed the health condition of urine donors.
Second: please tell me according to which scientific protocol you performed the processing of urine samples.
(Please describe both mentioned protocol in a supplementary file)
9. About the part "Results" in page 6-14
All over this part, sentences with reference should be ommitted or transfered to the part discussion.
Note: in this part, you should tell only about your results. Other data about comparison of your findings with other studies or the interpretation that originated from results should be mentioned in the part "Discussion" not result.
Please exert mentioned note to the part
"Results" in page 6-14, there are some sentences (especially with reference(s) at the end) that should be reconsidered or ommitted or translocated to the part "Discussion". If not, please explain why?
10. About the part "Discussion" in page 14
Please categorize your findings based on the most important to least important and after that, turn them to subheadings and discuss about them one by one. (Reconsider the part "Discussion" according to mentioned note)
11. Please check and adjust the "Reference list" based on the regulations of reference list of journal. (Titles, doi, the name of journal and ... )
Reviewer 3 Report
Dear Authors,
The manuscript entitled “Extracellular Vesicles isolation from large volume samples by PDMS-free microfluidic device” developed a protocol to isolate extracellular vesicles from samples with large volume, including urine and cell culture media. The authors compared their newly invented device and workflow to isolate EV from urine and cell culture media with the commonly used method used for EV isolation, including UC and SEC. This is an interesting study, however, the device reported in this manuscript has been published by the same group in another journal. It is well known that ultrafiltration can be used to concentrate sample from large volume. Besides, grammatical mistakes are throughout the manuscript. This manuscript does not meet the high standard of IJMS in the current form. I suggest to reconsider after major revision.
Comments and suggestions
1. The full name of COC-OSTE needs to be clarified in the abstract.
2. Line 16, compare to needs to be changed to compare with.
3. Line 21, which is able to.
4. Line 21, EVs or EV? Please be consistent throughout the manuscript.
5. Line 22, large volume samples.
6. Line 22, it is potential to be.
7. Line 30-31, how exosomes are formed should be further described.
8. Line 38, allow them to. Please delete for after allow.
9. Line 39-40, please rephrase the sentence.
10. Line 47-49, EV isolated by SEC are also contaminated by the particles which have similar size with EV, for instance, low density lipoprotein particles and chylomicrons. These should be mentioned.
11. Line 54, microfluidics have, not has.
12. Line 95, delete therefore.
13. Page 4, line 15, cells were cultured in, not on.
14. Page 4, Line 16, a weird symbol should be deleted after 4. Please also check throughout the manuscript.
15. Page 4, line 16, why large vesicles were not removed when they isolate EV from cell lines?
16. Page 4, line 17, to isolate EV from media, normally 2 round 100.000 g spinning is used. Could you please explain why you wash the UC collected EV 2 times?
17. Page 4, line 18, to concentrate samples, normally centrifuge 15-30 min is used. Could the authors explain why spin 1 hour?
18. Figure 4d, it seems that SEC or UC is much better to enrich EV from urine samples. Did the authors calibrate the samples to the same volume? If so, could the authors explain why more particles are observed in SEC and UC isolated samples?
19. Page 10, line 42, delete a period.
20. Page 11, line 43, cell culture media is far from complex compare with biological fluids, such as urine and blood. Please rephrase these sentences.
21. Figure 5h, why we can see much more EV in UC or SEC isolated samples?
22. Although the authors did ELISA to detect EV marker, it is not quite enough. To compare the isolation efficiency of different method, western blotting to detect EV specific marker is needed.
23. If doable, the authors may need to perform proteome analysis on the EV isolated using different methods. The methods optimized in this study may specifically enriched subtype of EVs, since it is a size and buoyancy based isolation method.
24. Please check type errors and grammar throughout the manuscript.
Round 2
Reviewer 1 Report
I suggest this manuscript can be accepted in present form.
Author Response
Dear reviewer,
Thank you for accepting the manuscript after the changes made. Your feedback and guidance were instrumental in improving the manuscript, and we are grateful for the opportunity to work with you on this.
Yours sincerely,
Cristina Bajo-Santos
Reviewer 3 Report
Dear authors,
Thank you for revising the manuscript entitled “Extracellular Vesicles isolation from large volume samples by PDMS-free microfluidic device”. Thank you for answering my comments point to point. However, I still have some comments/suggestions about this manuscript. Please see them below. I suggest to reconsider this manuscript after major revision.
Comments
1. What is PDMS should be clarified in the title.
2. Page 2, line 87, EVs or EV? Please check the manuscript thoroughly.
3. Page 2, line 88, change EV separation to EV isolation.
4. The less commonly used methods the authors described here is also commonly used. These methods are purification methods, not only used for EV isolation. Please rephrase it.
5. Please merge line 209-213 and line 214-219 to one paragraph.
6. Line 294, why these two cell lines were selected?
7. Please rephrase section 2.3 cell cultures. It may make readers confused.
8. Line 470, how A4F can be specifically used for exosome isolation? There is currently no such a kind of method that can totally separate exosome from microvesicles. Some microvesicles also have small size.
9. In figure 2C, the authors show the particle size of EV isolated using different methods. Please also show the peak plots for size distribution. The same for figure 3.
10. Can the method described in this manuscript efficiently remove contamination from clinical samples? For example, remove THP from urinary EV.
11. The legends of figures 4 and 5 are poorly written. Please add more details here.
12. Line 841, which gold standard method? UC?
13. Western blotting is a powerful way to show whether your samples contain CD63+ EV. If possible, please compare the level of CD63+ (or CD9, CD81) in the EV isolated using different methods. THP may also need to be detected using western blotting or ELISA, to further confirm that EV isolated using the newly developed method is highly purified.
14. Please further check type errors and grammar throughout the manuscript.
Author Response
Please, see the attachment

Round 3
Reviewer 3 Report
Dear authors,
Thank you for revising the manuscript. You have clearly answered my questions. This manuscript can be accepted in the current form.
Regards